# Serum β-hCG as a Biomarker in Pancreatic Neuroendocrine Tumors: Rethinking Single-Analyte Approach

**DOI:** 10.3390/cancers16112060

**Published:** 2024-05-29

**Authors:** Paweł Komarnicki, Paweł Gut, Maja Cieślewicz, Jan Musiałkiewicz, Adam Maciejewski, Michalina Czupińska, George Mastorakos, Marek Ruchała

**Affiliations:** 1Department of Endocrinology, Metabolism and Internal Diseases, Poznan University of Medical Sciences, Przybyszewskiego 49, 60-355 Poznań, Polandjan.musialkewicz@usk.poznan.pl (J.M.);; 2Unit of Endocrinology, Diabetes Mellitus and Metabolism, Aretaieion University Hospital, Medical School, National and Kapodistrian University of Athens, 157 72 Athens, Greece

**Keywords:** neuroendocrine tumors, tumor biomarkers, pancreatic neoplasms, chorionic gonadotropin, chorionic gonadotropin, beta subunit, human

## Abstract

**Simple Summary:**

The lack of biomarkers in neuroendocrine tumors (NETs) represents a critical unmet need. Conventional single analytes lack the required diagnostic accuracy. Multianalyte assays (e.g., NETest) offer improved performance but remain costly and largely unavailable. In our study, serum beta-human chorionic gonadotropin (β-hCG) shows promise as a biomarker in metastatic pancreatic NETs. β-hCG concentrations strongly correlate with established indicators of aggressive tumor behavior, including progressive disease, per RECIST; greater liver tumor burden; and higher WHO tumor grade. Throughout the study period, β-hCG levels increased consistently, with sharper rises noted in more aggressive tumors. As a widely available and inexpensive assay, serum β-hCG emerges as a useful tool for the evaluation of treatment response and disease monitoring in pancreatic NETs, especially in the transitional period until multianalyte assays become a mainstay in scientific guidelines.

**Abstract:**

Despite recent advances, neuroendocrine tumors (NETs) remain a challenging topic, due to their diversity and the lack of suitable biomarkers. Multianalyte assays and the shift to an omics-based approach improve on the conventional single-analyte strategy, albeit with their own drawbacks. We explored the potential of serum β-hCG as a biomarker for NETs and discussed its role in disease monitoring. We recruited 40 patients with non-functioning pancreatic NETs, all with liver metastases. Serum β-hCG concentrations were measured at 3-month intervals over 48 months. We performed a comparative and a repeated measures analysis of β-hCG depending on WHO grade (G1, G2), liver tumor burden (LTB; below 10%, 10–25%), and RECIST 1.1. (stable disease, progressive disease). Patients with progressive disease (*p* < 0.001), 10–25% LTB (*p* < 0.001) and WHO Grade 2 (*p* < 0.001) displayed higher β-hCG concentrations. Throughout the study, β-hCG concentrations consistently increased across the entire cohort. Delta β-hCG during the study period was greater in patients with 10–25% LTB (*p* < 0.001), progressive disease (*p* < 0.001), and G2 (*p* = 0.003). Serum β-hCG correlates with established indicators of malignancy and disease progression in metastatic NETs, supporting further studies as a monitoring and prognostic biomarker. Despite promising results from novel biomarkers, there is still a place for single-analyte assays in NETs.

## 1. Introduction

Neuroendocrine tumors (NETs) are a challenging topic for clinicians and researchers. They often stay undetected for extended periods, resulting in delayed diagnosis. This is attributed to the typically slow growth pattern of NETs and the frequently encountered absence of specific symptoms. Notably, 12–22% of patients are diagnosed with the disease already at metastatic stage, resulting in worse outcomes and compromised quality of life [1]. Early detection and the identification of high-risk groups are crucial for the improvement of patient care. As highlighted by the expert consensus statement, the lack of suitable biomarkers for diagnosis and monitoring remains a critical unmet need in the management of NETs [2].

Various molecules, including chromogranin A (CgA), 5-hydroxyindoleacetic acid (5-HIAA), N-terminal Prohormone of Brain Natriuretic Peptide (NT-proBNP), and Pancreatic Polypeptide (PP) have been studied as potential biomarkers in NETs [3,4,5]. The conventional monoanalyte-based approach (e.g., the aforementioned use of CgA) entails limited sensitivity and specificity [6,7]. An alternative strategy involves multianalytes, exemplified by the mRNA-based NETest. The NETest offers superior diagnostic parameters. However, it remains costly and largely unavailable—drawbacks typical for novel technologies. These issues are expected to diminish with broader adoption; nevertheless, the search for optimal solutions continues [7,8,9]. Furthermore, recent studies discussed potential prognostic utility of mutations in alpha-thalassemia/mental retardation X-linked chromatin remodeler (ATRX) and death-domain-associated protein (DAXX) genes in metastatic disease in pancreatic NETs. The loss of expression in ATRX/DAXX genes, often co-existing with the presence of Alternate Lengthening of Telomeres (ALT) correlates with more aggressive disease. Furthermore, research suggests the status of ATRX/DAXX and ALT as separate, negative prognostic biomarkers for pancreatic primary location of NETs [10,11].

Navigating the intricacies of biomarker research and its integration into clinical practice can be reminiscent of drinking from a firehose, as aptly described by Dunn et al. [12]. While multiplex assays are considered to be the future, their complexity is followed by a gradual introduction into routine care. In the interim, monoanalytes, despite their own drawbacks, offer a more readily available solution during this transitional period. In the light of these challenges, free beta human chorionic gonadotropin (β-hCG) emerges as a potential biomarker candidate. Commonly used for the detection of pregnancy and certain malignancies (e.g., choriocarcinoma), it could offer a relatively quick, inexpensive, and widely available measurement.

As per the WHO International Nonproprietary Names (INN) program, proteins with the same amino acid sequence share a common name. For this reason, the term human chorionic gonadotropin (hCG) refers to at least five distinct hCG forms, each produced by different cells, with unique structures and various functions [13,14]. These forms include hCG produced by syncytiotrophoblast cells in the placenta, sulphated hCG originating in the pituitary, hyperglycosylated hCG synthesized in cytotrophoblast cells, β-hCG, and hyperglycosylated β-hCG found in neoplasms. hCG and sulphated hCG attach to the hCG/luteinizing hormone (LH) receptor, whereas hyperglycosylated hCG, β-hCG, and hyperglycosylated β-hCG bind the transforming growth factor beta (TGF-β) receptor in an autocrine way [13]. hCG and hyperglycosylated hCG are the key factors during gestation. They are responsible for the implantation, uterine invasion, the creation of hemochorial placentation, and the maintenance of the corpus luteum. Their rising concentrations serve as a biomarker for the detection of pregnancy. Sulphated hCG co-promotes ovulation alongside LH and supplements its actions. By antagonizing the TGF-β receptor, hyperglycosylated hCG, β-hCG, and hyperglycosylated β-hCG block apoptosis and promote cell growth [13]. Hyperglycosylated hCG is involved in the development of choriocarcinoma and germ cell tumors, whereas β-hCG and hyperglycosylated β-hCG have been found in a variety of advanced neoplasms and are seemingly linked with disease progression [15,16,17,18,19,20]. Tissue expression and elevated serum concentrations of β-hCG have also been observed in NETs, with β-hCG considered a marker of poor prognosis. However, existing data are limited, and some findings are contradictory [4,18,21,22,23].

In the pursuit of novel biomarkers, we expect β-hCG to be a useful, widely available tool in monitoring advanced pancreatic NETs. We hypothesize that elevated serum concentrations of β-hCG will correlate positively with known indicators of malignancy, helping to predict disease progression.

## 2. Materials and Methods

### 2.1. Study Population and Design

This was a prospective study conducted over a 48-month period at the Department of Endocrinology, Metabolism, and Internal Diseases in Poznań, Poland. We recruited 40 patients with histopathological confirmation of non-functioning pancreatic neuroendocrine tumors (panNETs) and confirmation of liver metastases via imaging modalities, including SPECT, PET, CT, and MRI. All patients were treated with long-acting somatostatin analogs (SSA) at the time of the inclusion and throughout the study period (octreotide, Sandostatin LAR 30 mg once every 4 weeks, Novartis, Basel, Switzerland; lanreotide, Somatuline Autogel 120 mg once every 4 weeks, Ipsen, Paris, France).

Exclusion criteria were medical history of other malignancies, neuroendocrine carcinomas (NECs), and primary NETs located outside the pancreas; grading other than G1 and G2 in pancreatic NETs; prior or ongoing concomitant cancer treatment, including everolimus, sunitinib, radionuclide treatment, and systemic chemotherapy; pregnancy and breastfeeding; lack of compliance with follow-up visits, SSA regimens or dosing different than stated above.

After recruitment, we evaluated patients based on the following variables: sex, age, disease status according to RECIST 1.1, liver tumor burden (LTB), Ki-67 proliferation index, tumor grading.

We defined LTB as the percentage of liver volume infiltrated by NET metastases. LTB was evaluated based on a contrast-enhanced CT scan of the abdomen performed at the time of the inclusion. Patients were divided into 2 subgroups: (a) those with LTB up to 10%, and (b) those with LTB between 10% and 25%. Consecutive CT scans were used to monitor disease status. Disease status was defined based on RECIST 1.1.and by comparing consecutive imaging to the baseline CT. Patients developing progressive disease (PD) over the study period and patients with stable disease (SD) were distinguished within the study group.

Ki-67 index was based on a histopathological evaluation of tumor tissue upon diagnosis. Grading was divided into G1 and G2 subgroups based on 2022 WHO criteria [24].

### 2.2. Data Collection and Assays

Data collection was performed during routine control visits at the outpatient clinic of the Department of Endocrinology on the days of SSA injection. Fasting venous blood samples were collected in the morning on the day of the visit, before SSA administration. After collection, samples were transferred to the laboratory unit at the hospital, where plasma was obtained by centrifugation. Plasma samples were analyzed within 6 h from collection. Serum β-hCG was measured every 3 months throughout the study period, acquiring 16 samples from each patient and 640 samples in total. Serum β-HCG concentrations were assessed by Elecsys HCG + β assay (Roche, Rotkreuz, Switzerland). The assay has an analytical measuring range between 0.2–10,000 mIU/mL. There were no results outside of the test’s measuring range in our analysis. The β-HCG reagent kit was used according to the manufacturer’s standardized protocol. Technicians performing the assays were blinded to the study endpoints. Samples were labeled with pseudonymized IDs, with no indicating clinical information provided.

Patients underwent routine CT scans every 6 months for disease evaluation.

### 2.3. Statistical Analysis

Baseline demographic and clinical characteristics were summarized using descriptive statistics. Distributions of continuous variables were assessed for normality using the Shapiro–Wilk test. For non-normally distributed continuous variables, differences between groups were analyzed using the Mann–Whitney *U* test with continuity correction. The Friedman test with Dunn–Bonferroni multiple comparison tests was applied for the analysis of non-normal β-hCG longitudinal data. Trends were assessed with the Page test. Normally distributed longitudinal β-hCG concentration datasets violating the sphericity assumption were analyzed using ANOVA Friedman test. Delta β-hCG was defined as the difference between the final and initial β-hCG measurements and was examined by Mann–Whitney *U* test with continuity correction, due to a lack of normal distribution within datasets. We evaluated correlations between continuous variables using Spearman’s rank correlation coefficient. Univariate logistic regression analysis was conducted to assess the value of initial β-hCG concentration in predicting treatment response. We generated a Receiver Operating Characteristic curve (ROC), and calculated the area under the curve (AUC), along with its 95% confidence intervals (CI), using DeLong’s method to assess the performance of the model.

All tests were two-tailed with the significance level set at α = 0.05. *p* values < 0.05 were considered statistically significant. Analysis was performed using Statistica version 13 (TIBCO Software Inc., Palo Alto, CA, USA), and PyCharm 2024 1.1. Integrated Development Environment (Jet Brains, Prague, Czech Republic), with data processing, modeling, and visualization performed using Python 3.12. programming language with the pandas 2.2.2, statsmodels 0.14.2, scikit-learn 1.4.2, numpy 2.0.0rc1, and matplotlib 3.8.4 packages.

## 3. Results

### 3.1. Baseline Characteristics

We included 40 non-functioning panNETs patients. Participants had a mean age of 63 and were predominantly female (70%). Of the study group, 45% had a G1 tumor, 47.5% had stable disease according to RECIST 1.1., and 55% had a LTB score of under 10%. Study group characteristics are displayed on Table 1.

### 3.2. β-hCG Concentration Depending on Tumor Malignancy

We examined the relationship between serum β-hCG concentrations and indicators of cell proliferation, including grading, treatment response, and LTB. Each patient underwent 16 β-hCG measurements at 3-month intervals. β-hCG showed a strong positive correlation with PD, G2, and 10–25% LTB. The level of statistical significance was <0.001 across all variables. Moreover, the correlation was consistent and persisted at all measuring time points. The results of this analysis are displayed in Table 2, Table 3 and Table 4. Complete statistical analysis is available in the Appendix A.

### 3.3. β-hCG Changes over Time

We observed a substantial increase in β-hCG concentrations across 16 measurements in the study group and in all subgroups. Moreover, the 95% confidence intervals within the subgroups did not overlap at any time point, as shown in Figure 1, Figure 2 and Figure 3. Complete results of the repeated measures analysis, including post hoc analysis and Page test for trend are provided in the Appendix A. Additionally, we assessed the relationship between delta β-hCG and the study variables. The increase in β-hCG was greater in patients with PD, 10–25% LTB, and G2. Age of the participants was not a significant factor in the analysis. The results of this analysis are shown in Table 5 and Figure 4, Figure 5 and Figure 6.

### 3.4. β-hCG and Predicting Treatment Response

We performed a logistic regression analysis to evaluate the association between treatment response and initial β-hCG concentration. The analysis revealed the initial β-hCG levels can predict the occurrence of PD within the study group (beta = 0.0171, OR= 1.0173, *p* = 0.003). The results of this analysis are presented in Table 6.

Furthermore, the ROC curve demonstrated good discriminative ability of the logistic regression model, with an AUC of 0.9173. The optimal sensitivity and specificity were 90.48% and 94.74% respectively. The results of ROC analysis are displayed in Figure 7.

## 4. Discussion

Our findings reveal a strong correlation between serum β-hCG concentrations and known indicators of aggressive panNET behavior, including disease progression per RECIST 1.1, high tumor grade, and large volume of tumor metastases. This correlation remained consistent throughout a 48-month follow-up period, suggesting potential usefulness of β-hCG as a monitoring biomarker. β-hCG has also shown good accuracy in predicting PD in metastatic panNETs.

### 4.1. History of β-hCG Research in NETs

The first evidence of aberrant β-hCG secretion in NETs was provided by Kahn et al. in 1977. Initially, elevated β-hCG concentrations were associated with the presence of panNETs, as opposed to the ectopic production of alpha hCG (a-hCG) in midgut NETs. Similar observations were made by Oberg et al. in their 1981 study [23,25]. In 1987, Heitz et al. examined tissue expression of hCG subunits in panNETs. The observation was not confirmatory for β-hCG, with the results attributed to the insufficient sensitivity of the testing method [18]. Over time, however, elevated serum β-hCG concentrations were detected in 12–30% of panNETs, supporting the existence of ectopic production in a substantial group of patients [21,22,26,27]. Notably, combining β-hCG measurement with chromogranin A or alpha-fetoprotein showed improved diagnostic accuracy and potential utility in identification of patients with poor prognosis [21]. Still, the data remain scarce and scientific guidelines recommend against routine use of β-hCG for diagnosis and monitoring [6,28,29,30].

### 4.2. Commentary on the Results of the Study

A statistical association between elevated β-hCG concentrations and variables indicating high levels of cellular proliferation support the hypothesis of its use as a marker of poor prognosis in NETs, as outlined by Shah et al. [21]. Higher β-hCG concentrations in the presence of PD, as per RECIST 1.1., are of particular interest, suggesting its potential as a prognostic biomarker. Moreover, a rising trend of β-hCG concentrations can be used for serial monitoring to alert physicians of potential progression in high-risk patients.

The results of the study should be interpreted in the context of different hCG forms, with distinct roles and structures. As noted in the introduction, hCG forms linked with the neoplastic process include hyperglycosylated hCG (secreted by the choriocarcinoma and germ cell tumors), β-hCG, and hyperglycosylated β-hCG (found in various neoplasms). It is hypothesized that as neoplasms undergo mutations and progress, they begin to produce β-hCG or hyperglycosylated β-hCG. These neoplasm-linked forms bind to the TGF-β receptor, inhibiting apoptosis [14,31]. We suggest that the same mechanism may occur in advanced NETs. As highlighted earlier, less than half of panNETs were found to produce β-hCG [21,22,26,27]. However, these studies included mostly patients with localized disease, while our research focused exclusively on subjects with metastatic disease. In this setting, a rising trend was present across all patients with metastases, with a sharper rise detected in the more aggressive tumors.

We align with the view of the expert consensus that, given a diverse nature of NETs, relying on a single substance as a solution to the lack of suitable biomarkers is unlikely [2]. Several researchers supported the addition of β-hCG to the already existing diagnostic algorithm in order to improve its diagnostic accuracy [6,7,21,32]. β-hCG measurement represents a widely available and relatively inexpensive option, especially when compared to complex multianalyte assays. Serial β-hCG monitoring could provide physicians with additional information about the state of the disease and help guide management decisions. A more accurate depiction of the disease acquired by combining β-hCG with other biomarkers and imaging modalities would help guide patient care in the transitional period until multianalytes become a routine tool. As an example, a substantial rise between measurements (e.g., doubling time) or an elevated value in a single sample, could serve as an indicator for the earlier introduction of another form of treatment. In this setting, a more thorough research of β-hCG in NETs is justified. Lastly, evaluating urinary hCG levels and comparing them with serum concentrations presents another interesting research direction. Qualitative and quantitative analysis of urinary β-hCG concentration could present an inexpensive and non-invasive alternative for monitoring of the disease status. However, quantitative β-hCG assays are not widely available and, while the qualitative tests are easily accessible, they provide little information and may not detect the specific β-hCG isoforms produced by NETs [33].

### 4.3. Study Limitations

Biomarkers in NETs (e.g., chromogranin A) typically have different diagnostic accuracy depending on the primary location of the tumor. Hence, the main limitation of our study remains the sole inclusion of pancreatic NETs. Furthermore, all patients throughout the study period were treated with SSA, with its impact on β-hCG being unknown. Even though β-hCG concentrations in patients with indicators of more malignant disease were higher across all time points, the time since the introduction of SSA in each patient differed and was not accounted for in the calculations. Even though we present a rationale for the use of β-hCG in the group of patients with advanced disease, our study does not provide an analysis for its use in early detection and localized disease, where the need for biomarkers is crucial. A comparison with non-metastatic panNETs and a healthy control group is required to evaluate its potential diagnostic utility. However, considering that β-hCG is usually expressed in more aggressive NETs, its role as a diagnostic biomarker could be limited. Given the single-center aspect of our trial and the rare disease nature of NETs, the limited sample size presents another limitation. As displayed in Figure 8, there was a notable overlap of variables, indicating an aggressive disease, where patients with PD often also presented with a G2 tumor and high LTB. Furthermore, the WHO Grade 2 patients in the study presented with Ki-67 levels up to 10%. Patients with Ki-67 levels in the 10–20% range were not accounted for, thus not representing the full spectrum of G2 patients. PanNETs with Ki-67 index over 10% often have similar disease characteristics to other G2 patients, however a number of clinical trials (such as CLARINET or NETTER-2) have separated these two subgroups [34]. Additionally, incorporating variables such as the status of ATRX/DAXX, ALT, and information about primary tumor size could enhance the analysis and provide a more comprehensive depiction of the disease status. However, these immunohistological markers are not yet routine and were not part of our analysis. While this observation highlights the correlation between aggressive tumor behavior and β-hCG concentration, it also underscores the presence of various compounding factors requiring consideration when interpreting study results. Moreover, elevated β-hCG is present in various neoplasms, as well as benign diseases, and in pregnancy, which can lead to false positive measurements [20].

Finally, the observed performance metrics of the logistic regression analysis raise concerns regarding potential model overfitting. High sensitivity and specificity in predicting the occurrence of PD are encouraging. However, the low *p*-value from the Hosmer–Lemeshow test (*p* = 0.0294, as shown in the Appendix A in the Appendix A) and the highly selective nature of the patient recruitment process discussed in this paragraph suggest there is a risk that the model has learned specific patterns from the training data that may not generalize well to new, unseen data. Further investigation and validation studies are warranted to assess the robustness and generalizability of the model.

## 5. Conclusions

The results of our research remain consistent with prior studies identifying links between elevated β-hCG in NETs and disease severity. Our initiative for further research and possible introduction of β-hCG assessment into routine is amplified by the continuously rising incidence rates of NETs worldwide and the remaining lack of suitable biomarkers [35,36]. Overcoming of scientific and economic barriers requires time. Multianalytes offer a promising, albeit relatively distant, vision of the future. In this context, we conclude that β-hCG warrants further investigation as a NET biomarker. We propose future directions of research to include studies across different disease characteristics (different primary locations, localized versus advanced disease), as well as focusing on changes over time depending on an outcome (e.g., β-hCG doubling time, change in β-hCG relative to applied treatment).

## Figures and Tables

**Figure 1 cancers-16-02060-f001:**
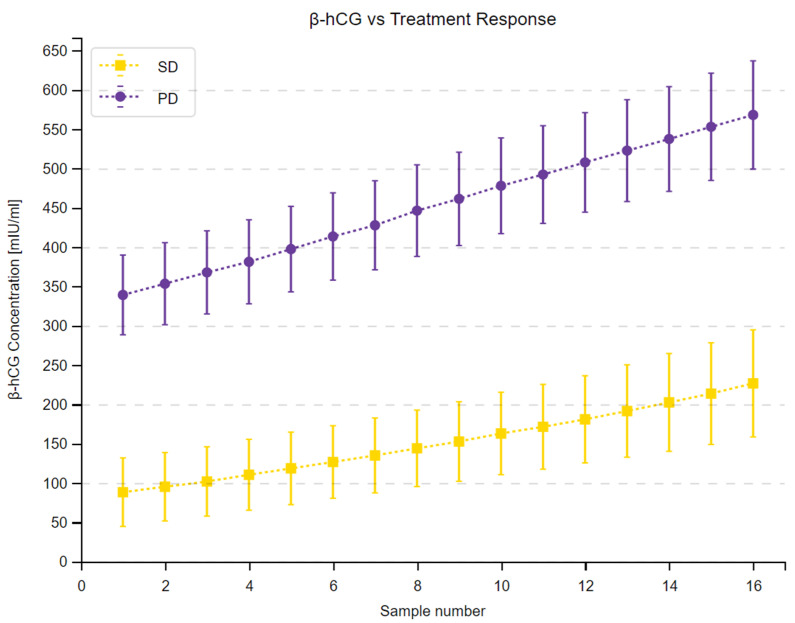
The changes in β-hCG concentration over the 48-month study period depending on the treatment response according to RECIST 1.1. Mean β-hCG concentrations at a given time point determined by a sample number are presented as squares in patients with stable disease (SD) and circles in patients with progressive disease (PD). Vertical lines display 95% confidence intervals in each of the time points.

**Figure 2 cancers-16-02060-f002:**
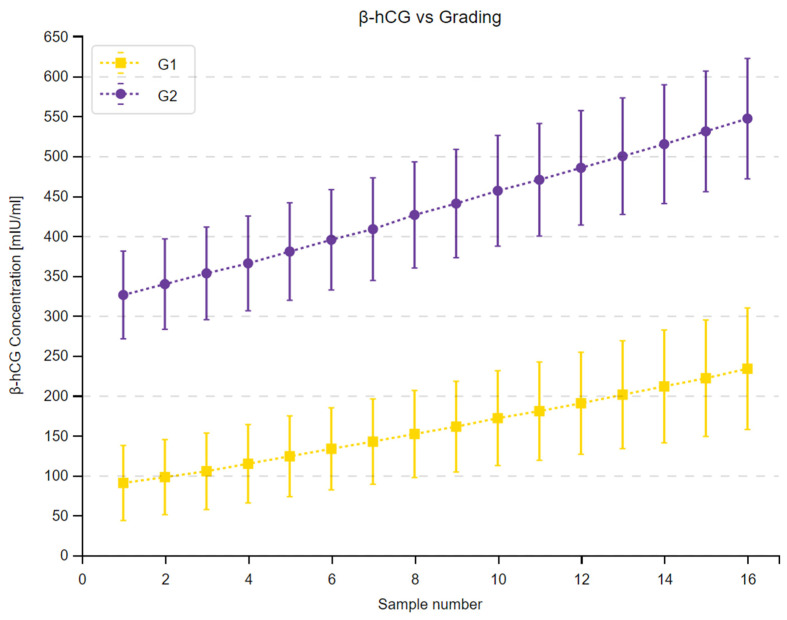
The changes in β-hCG concentration over the study period depending on tumor grading based on 2022 WHO criteria. Mean β-hCG concentrations at a given time point determined by a sample number are presented as squares in patients with G1 tumors and circles in patients with G2 tumors. Vertical lines display 95% confidence intervals in each of the time points.

**Figure 3 cancers-16-02060-f003:**
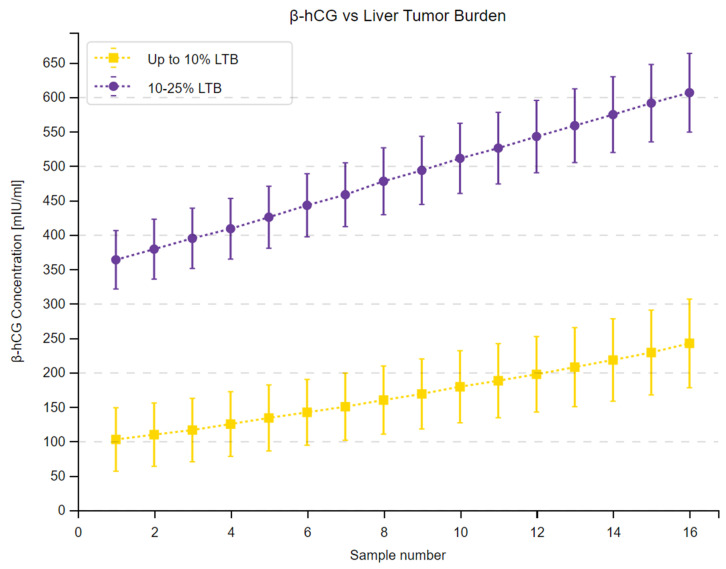
The changes in β-hCG concentration over the 48-month study period in subgroups depending on liver tumor burden (LTB). Mean β-hCG concentrations at a given time point determined by a sample number are presented as squares in patients with LTB up to 10% and circles in patients with LTB between 10 and 25%. Vertical lines display 95% confidence intervals in each of the time points.

**Figure 4 cancers-16-02060-f004:**
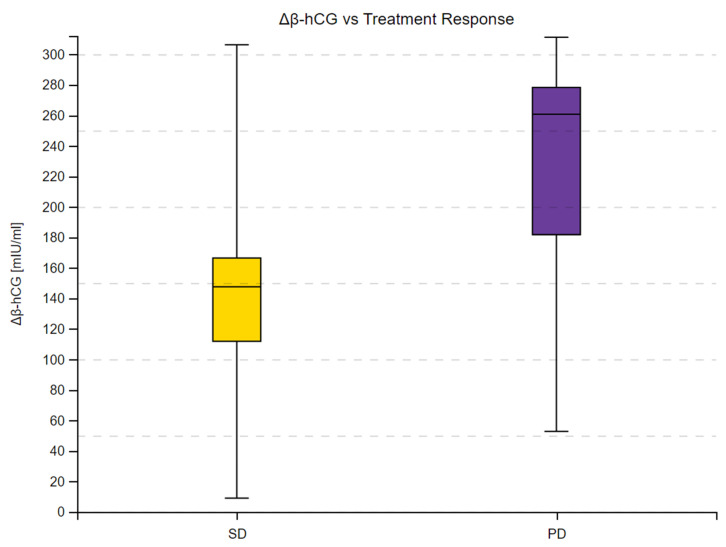
Delta β-hCG concentration in patients with stable disease (SD) and progressive disease (PD) according to RECIST 1.1. Delta β-hCG represents the individual variation between final and initial β-hCG concentrations (measured in mIU/mL). Box plots depict the interquartile range (Q1 to Q3), with the median highlighted by the internal line. The whiskers extend to the minimum and maximum values of the dataset.

**Figure 5 cancers-16-02060-f005:**
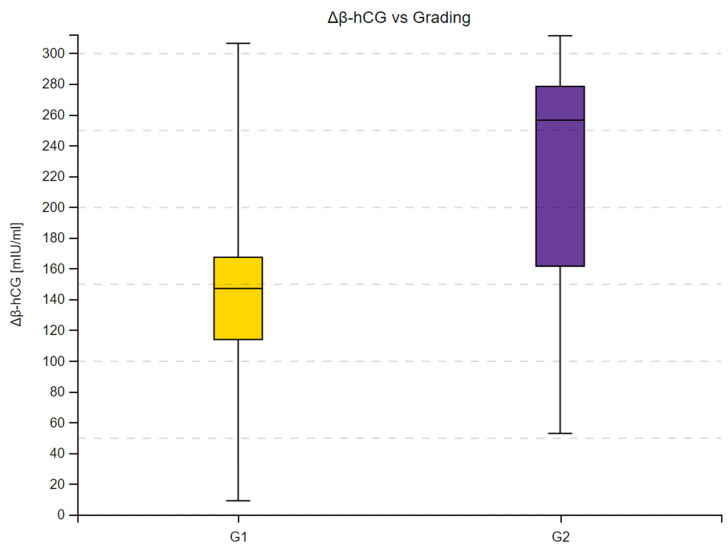
Delta β-hCG concentration in patients with G1 and G2 tumors defined by 2022 WHO grading. Delta β-hCG represents the individual variation between final and initial β-hCG concentrations (measured in mIU/mL). Box plots depict the interquartile range (Q1 to Q3), with the median highlighted by the internal line. The whiskers extend to the minimum and maximum values of the dataset.

**Figure 6 cancers-16-02060-f006:**
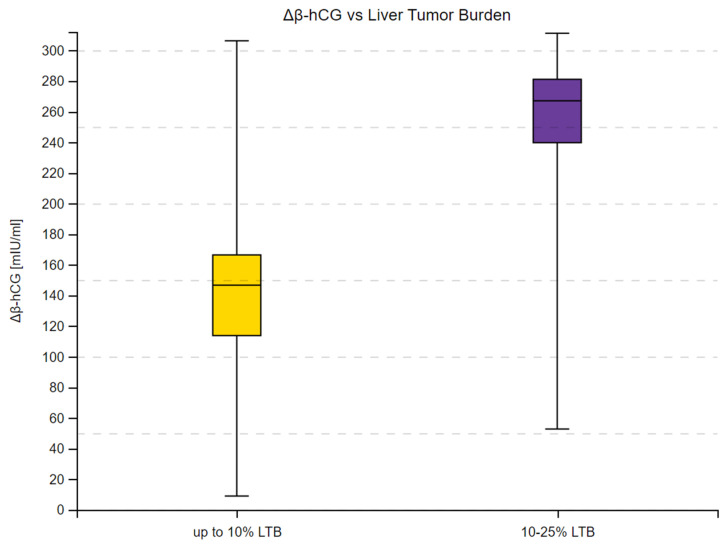
Delta β-hCG concentration in patients with up to 10% and 10–25% liver tumor burden (LTB). Delta β-hCG represents the individual variation between final and initial β-hCG concentrations (measured in mIU/mL). Box plots depict the interquartile range (Q1 to Q3), with the median highlighted by the internal line. The whiskers extend to the minimum and maximum values of the dataset.

**Figure 7 cancers-16-02060-f007:**
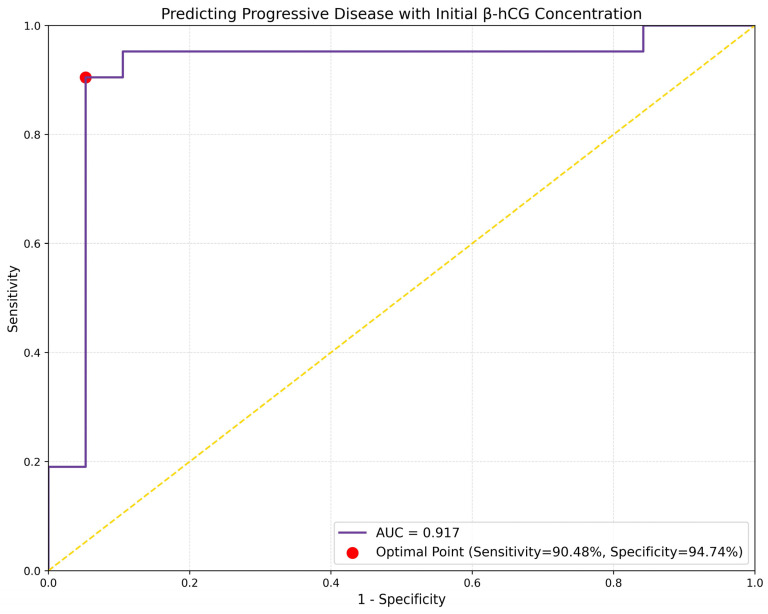
ROC curve analysis illustrating the prognostic accuracy of initial β-hCG concentration on the occurrence of progressive disease. The dotted line represents the ROC curve if the classification is randomly estimated. The red dot indicates the optimal operating point for sensitivity and specificity.

**Figure 8 cancers-16-02060-f008:**
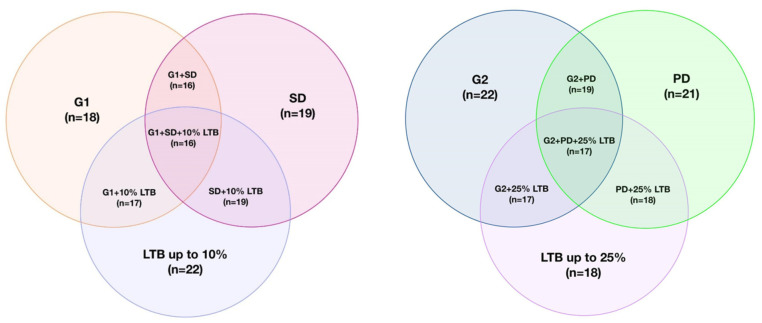
Gantt diagrams illustrating study group characteristics and the overlap between variables in the study subjects.

**Table 1 cancers-16-02060-t001:** Baseline characteristics of the study group.

Variable	All Patients (*n* = 40)
**Demographics**	
Age, years	63 (8) *
**Sex**	
Male	12 (30%)
Female	28 (70%)
**Clinical characteristics**	
**Primary tumor location**	
Pancreas	40 (100%)
**Liver Tumor Burden**	
Up to 10%	22 (55%)
10–25%	18 (45%)
**Grading by WHO**	
G1	18 (45%)
G2	22 (55%)
**Treatment response by RECIST 1.1.**	
Stable Disease (SD)	19 (47.5%)
Progressive Disease (PD)	21 (52.5%)
**Ki 67**	
1%	2 (5%)
2%	16 (40%)
4%	2 (5%)
5%	8 (20%)
10%	12 (30%)

* Mean (S.D.).

**Table 2 cancers-16-02060-t002:** Relationship between the serum concentration of β-hCG and treatment response according to RECIST 1.1 across 16 evenly distanced measurements over 48 months.

Measuring Time Point	Median β-hCG—SD [mIU/mL]	Median β-hCG—PD [mIU/mL]	Sum of Ranks SD	Sum of Ranks PD	*U*	Z Corrected	*p*
1.	67.80	356.40	223	597	33	−4.4964	<0.001
2.	76.50	376.30	220	600	30	−4.5783	<0.001
3.	86.30	389.50	219	601	29	−4.6047	<0.001
4.	93.40	402.60	220	600	30	−4.5776	<0.001
5.	102.30	420.40	219	601	29	−4.6047	<0.001
6.	113.40	438.40	218	602	28	−4.6314	<0.001
7.	123.40	451.20	219	601	29	−4.6045	<0.001
8.	134.20	462.40	217	603	27	−4.6587	<0.001
9.	143.50	476.50	218	602	28	−4.6314	<0.001
10.	153.40	487.60	219	601	29	−4.6043	<0.001
11.	159.40	496.70	219	601	29	−4.6043	<0.001
12.	167.80	515.80	220	600	30	−4.5772	<0.001
13.	178.50	527.80	222	598	32	−4.5230	<0.001
14.	189.70	546.70	223	597	33	−4.4964	<0.001
15.	199.60	576.30	223	597	33	−4.4962	<0.001
16.	212.80	598.70	223	597	33	−4.4959	<0.001

**Table 3 cancers-16-02060-t003:** Relationship between the serum concentration of β-hCG and grading according to RECIST 1.1 across 16 evenly distanced measurements over 48 months.

Measuring Time Point	Median β-hCG—G1 [mIU/mL]	Median β-hCG—G2 [mIU/mL]	Sum of Ranks G1	Sum of Ranks G2	*U*	Z Corrected	*p*
1.	67.70	354.35	211	609	40	−4.2822	<0.001
2.	76.50	368.75	211	609	40	−4.2829	<0.001
3.	85.40	375.70	210	610	39	−4.3094	<0.001
4.	95.60	390.60	213.5	606.5	42	−4.2143	<0.001
5.	102.40	409.55	214	606	43	−4.2007	<0.001
6.	114.40	427.60	213	607	42	−4.2275	<0.001
7.	123.45	442.00	216	604	45	−4.1461	<0.001
8.	135.45	457.80	215	605	44	−4.1733	<0.001
9.	143.60	473.15	217	603	46	−4.1187	<0.001
10.	154.90	480.40	216	604	45	−4.1459	<0.001
11.	162.55	488.95	216	604	45	−4.1459	<0.001
12.	171.05	507.25	215	605	44	−4.1731	<0.001
13.	182.95	522.15	216	604	45	−4.1459	<0.001
14.	192.55	545.20	217	603	46	−4.1191	<0.001
15.	200.95	570.35	218	602	47	−4.0917	<0.001
16.	213.10	593.15	216	604	45	−4.1459	<0.001

**Table 4 cancers-16-02060-t004:** Relationship between the serum concentration of β-hCG and liver tumor burden according to RECIST 1.1 across 16 evenly distanced measurements over 48 months.

Measuring Time Point	Median β-hCG—up to 10% [mIU/mL]	Median β-hCG—10–25% [mIU/mL]	Sum of Ranks up to 10%	Sum of Ranks 10–25%	*U*	Z Corrected	*p*
1.	72.15	382.20	275	545	22	−4.7717	<0.001
2.	79.40	397.15	272	548	19	−4.8539	<0.001
3.	87.40	413.05	271	549	18	−4.8804	<0.001
4.	95.60	429.95	271	549	18	−4.8804	<0.001
5.	102.40	442.80	271	549	18	−4.8804	<0.001
6.	114.40	457.90	270	550	17	−4.9071	<0.001
7.	123.45	472.40	271	549	18	−4.8802	<0.001
8.	135.45	490.95	268	552	15	−4.9617	<0.001
9.	143.60	510.25	269	551	16	−4.9343	<0.001
10.	154.90	523.00	269	551	16	−4.9343	<0.001
11.	162.55	535.05	269	551	16	−4.9343	<0.001
12.	171.05	550.30	269	551	16	−4.9343	<0.001
13.	182.95	562.30	271	549	18	−4.8800	<0.001
14.	192.55	578.30	271	549	18	−4.8804	<0.001
15.	200.95	594.70	272	548	19	−4.8530	<0.001
16.	213.10	612.15	272	548	19	−4.8530	<0.001

**Table 5 cancers-16-02060-t005:** Relationship between delta β-hCG and study variables: treatment response, grading, liver tumor burden, and patient age. Mann–Whitney *U* tests were employed for treatment response, grading, and liver tumor burden, while the Spearman rank correlation test was used for patient age.

Variable		Median β-hCG [mIU/mL]	Sum of Ranks	*U*	Z Corrected	*p*
**Treatment Response**						
	SD	147.50	263	73	−3.4127	<0.001
	PD	260.70	557			
**Grading**						
	G1	143.23	261	90	−2.9227	0.003
	G2	256.25	559			
**Liver Tumor Burden**						
	up to 10%	146.65	307	54	−3.9014	<0.001
	10–25%	267.00	513			
**Patient Age**		**R Spearman**		**T (N-2)**		0.97
		−0.0069		−0.0423		

**Table 6 cancers-16-02060-t006:** Logistic regression analysis of initial β-hCG concentration for predicting progressive disease.

Variable	Beta Coefficient	95% CI for Beta	*p*-Value	Odds Ratio	95% CI for OR
Intercept	−3.2841	(−5.2093, −1.359)	0.0008	0.0375	(0.0055, 0.2569)
Initial β-hCG	0.0171	(0.0079, 0.0264)	0.0003	1.0173	(1.0079, 1.0268)

## Data Availability

The data presented in this study are available from the corresponding author upon request.

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
