# Peer review of "Serum β-hCG as a Biomarker in Pancreatic Neuroendocrine Tumors: Rethinking Single-Analyte Approach"

_cancers, 2024, doi:10.3390/cancers16112060_

Round 1
Reviewer 1 Report
Comments and Suggestions for Authors
The Authors investigate the potential role of beta-Hcg as biomarker for follow-up of PanNETs G1-G2 treated with standard dose SSAs. The study is well-designed, prospective, and the need for new and affordable biomarkers for these neoplasms is very strong.
Major comments:
- kaplan-meier+cox-regression analysis, or at least logistic regression to correlate increase in beta-hcg and occurrence of disease progression is needed in order to assess the ability of this marker in predicting progression
Minor comments:
- line 122: please, add the reference for WHO classification
Author Response
Dear Reviewer, thank you for your positive comments and the suggestions regarding our manuscript. We appreciate the time and effort you have dedicated to providing a thorough review of our work. We made several changes to the manuscript and we submit our responses below.
Major Comment:
Regarding your suggestion to improve the statistical aspect of our study and to evaluate the increase in beta-hCG with the occurrence of disease progression, we have performed a logistic regression analysis. In our opinion this method is more suitable for our dataset, compared to the Cox Regression and Kaplan Meier. It is better suited to a comparison of binary outcome variables (such as in our dataset - SD/PD), instead of survival analysis (time-to-event), in which the latter two excel.
In our corrected manuscript we evaluated the relationship between the initial β-hCG concentration and the occurrence of Progressive Disease (PD) within the study group. Our rationale for using the initial β-hCG concentration, instead of the rise in b-hCG (as was suggested in the first round of peer review) was that if b-hCG could predict PD, it would be more valuable for clinical decision-making (the clinical decisions could be made before PD is found on CT). In contrast, using the delta β-hCG could be potentially less useful, as the disease might have already progressed by the time the increase is detected.
The results of the logistic regression analysis are promising, however, they are impacted by the limitations of our study - the highly selective nature of patient recruitment and the low p-value from the Hosmer-Lemeshow test. This indicates a possible overfitting of the model, and we also highlighted this issue in the Limitations section. Despite these issues, after addressing the caveats, we decided to include it in the manuscript based on your suggestions.
Minor Comment:
We have added the reference for the 2022 WHO classification, as requested.
We thank you again for your valuable feedback and hope that our revisions and additional analyses have addressed your concerns. Please let us know if you have any further comments or suggestions.
Best regards, Paweł Komarnicki, on behalf of all co-authors
Reviewer 2 Report
Comments and Suggestions for Authors
The study is interesting with clear results but in some way limited. The case design and patients are with particular situation, grade NETG1 and NETG2, with liver metastasis ( in TNM classification M1a) , the immunohistochemical analyses is just for Ki67 with 1,2% categories means G1 and 4,5,10% categories means G2. From 10% to 20% still NETG2 we don't have patients evaluated. We still have criteria about size of tumor and immunohistochemical markers specific DAXX, ATRX , ALT for prognostic in pancreatic NET tumors.
Author Response
Dear Reviewer,
Thank you for your comments and the critique regarding our manuscript. We appreciate you highlighting the limitations of our study design. I made changes to the manuscript and I submit my response below:
We agree that our study focused on a specific subset of patients with pancreatic NETs - all metastatic to the liver and all with Ki-67 up to 10% (so G1 and lower Ki-67 G2), which might present a limitation. We adhered to your comment and discussed this in the limitation section.
We also want to highlight, that G2 up to 10% is a common cutoff point in NET studies and even clinical trials, as seen in the CLARINET and NETTER-2. Furthermore, we have seen that despite this limitation, there was a marked difference between hCG concentrations in G2 and G1 subgroups - with the whole spectrum of G2 patients we expect the difference to be even wider (although this remains speculation, with no statistical analysis to support it - also mentioned as the limitation).
We agree with your comment regarding the additional criteria for prognosis in pancreatic NETs, and the use of new immunohistochemical markers. We think that this aspect is promising, as evidenced by recent publications and sessions dedicated to this topic at this years ENETS. However, we did not incorporate these markers in our study, as they were not routinely analyzed in our department at the time this study was initiated more than 4 years ago, and they are not yet part of our standardized procedure. However, we have acknowledged the potential prognostic utility of these markers in the Introduction and Limitations sections of our manuscript and we think that they are an interesting subject for future research.
We thank you for your valuable feedback and the opportunity to address these concerns. We hope that our changes address all the issues highlighted in the first round of peer review and we are looking forward to your response.
Best regards,
Paweł Komarnicki, on behalf of all co-authors
Reviewer 3 Report
Comments and Suggestions for Authors
Dear Authors,
This article presents very interesting findings on the potential of serum beta-hCG for monitoring and prognostication in metastatic pancreatic NET. The article is well-written and the study is well-conducted, so I will recommend it to the Editor-in-Chief for publication, however there are 3 minor points that I recommend clarifying:
1) It should be made clear from the title, abstract and introduction that you are investigating only "metastatic pancreatic NETs" as opposed to "NETs".
2) Have you thought of using urinary hCG tests? Although quantitative urinary hCG tests are not that widespread, they could provide a non-invasive "screening" method for patients to monitor their own disease at home. If this has been done before, you should mention it, otherwise you should propose it as future direction of this research field.
3) A comparison with patients with non-metastatic pancreatic NETs is needed, and it could provide a useful "cut-off" value (and therefore sensitivity and specificity values), as long as information on the incidence of beta-hCG production in either group. If this has been done before, you should mention it, otherwise you should propose it as future direction of this research field.
Author Response
Dear Reviewer,
Thank you for your positive comments and the feedback on our article. We appreciate your recommendation for publication and the opportunity to clarify the points you raised.
1 - We have made changes to better illustrate the patient group that we study as these results might not necessarily translate to other locations.
2) Regarding the use of urinary hCG tests, we acknowledge that this is an interesting perspective for future research. Urinary tests can detect certain neoplasms and, should b-hCG find use as a biomarker in NETs, they could prove useful in a similar way. While quantitative urinary hCG tests are not widely available and were not considered in our study, we recognize the potential value of a non-invasive monitoring. We have discussed this as a potential future direction in the revised manuscript, while addressing the potential limitations of urinary tests.
3) We agree that establishing a cut-off value for β-hCG would greatly improve its utility. However, such analysi would be best suited in comparison with patients who have non-metastatic pancreatic NETs or versus healthy patients. Our current study focused on a selective group of patients with metastatic disease, which may not be suitable for deriving a reliable cut-off value. As you suggested, this has not been extensively investigated before, and we have proposed it as a future direction for research in this field.
We thank you for your valuable feedback and for the opportunity to clarify these points. Your suggestions have helped strengthen our manuscript, and we appreciate your contribution to our article.
Best regards,
Paweł Komarnicki, on behalf of all co-authors
Round 2
Reviewer 1 Report
Comments and Suggestions for Authors
I consider the paper adequate for publication in the present form